# The Impact of Transportation on the Croatian Economy: The Input–Output Approach

**Luka Vukić** [1,*], **Davor Mikulić** [2] and **Damira Keček** [3]

1    Department for Maritime Management Technologies, Faculty of Maritime Studies, University of Split, Ruđera Boškovića 37, 21000 Split, Croatia
2    The Institute of Economics Zagreb, Trg J. F. Kennedyja 7, 10000 Zagreb, Croatia; dmikulic@eizg.hr
3    University Center Varaždin, University North, 104. brigade 1, 42000 Varaždin, Croatia; dkecek@unin.hr
*    Correspondence: lvukic@pfst.hr

**Abstract:** The aim of this paper was to determine the economic impact of the transportation sector on the Croatian economy by using input–output analysis. According to the input–output tables for the Croatian economy for 2004, 2010, 2013, and 2015, output and gross value-added multipliers were calculated. The results of the conducted analysis indicated that the multiplicative effects of the transportation sector in Croatia were significant in the observed period, especially for the air transport sector. Furthermore, comparative multiplier analysis with selected European Union countries was performed to assess the Croatian transportation industry position from an international perspective. Lower output and gross value-added multipliers for the Croatian transportation sector imply that old European Union member states capitalized the transportation sector more for growth and development. The Croatian transportation sector recorded lower imported intermediate inputs, average domestic inputs, and higher value-added multipliers similar to new European Union members. Simulations based on multiplicative effects show that restrictions on movements and human contacts, imposed due to the COVID-19 pandemic, could induce a strong reduction in the economic activity of transport and other sectors that are included in the value-added chain of the transport industry.

**Keywords:** transportation sector; input–output analysis; multipliers; Croatia; European Union

**JEL Classification:** C67; R40



## 1. Introduction

Transportation is an essential link for the movement of individuals and goods. It undoubtedly contributes to the development of all business sectors and society. Transport plays an important role in the operation of each economy and is considered to be a determining factor for economic development and growth. The role of transportation is visible for its contribution towards the creation of an effective connection of the supply chain of goods and services, shipments of intermediate inputs, and delivery of final goods. The interrelationship between transport and other economic sectors has been widely examined by both the business and academic community from the macroeconomic and microeconomic perspectives. From the microeconomic perspective, the importance is usually assessed by its influence on each specific sector in the economy. The significance of the transport sector on the macroeconomic level is manifested in the overall impact to the output, income, economic growth, and employment. According to previous studies, the transport industry provides more than 10 million jobs in the transport industry and accounts for more than 5% of the overall gross domestic product (GDP) in the European Union (European Commission 2020a). An even greater share of the transport industry contribution should be indicated for developed European countries, varying between 6% and 12% of the GDP (Gnap et al. 2018). The transportation sector plays an important role

in the Croatian economy, accounting for 5% of the total GDP (CBS 2020b) and generating a significant share of 5.2% of total exports (The World Bank 2020). The importance of the transport sector in the Croatian economy (Božičević et al. 2008) is relatively higher than in other economies (Lejour et al. 2009).

The provision of transport and warehousing services requires a considerable capital investment (Rašić-Bakarić 2013), especially investments in transport infrastructure, which are considered to be essential for economic and social development (Ministry of the Sea, Transport, and Infrastructure of the Republic of Croatia 2017). The importance of the transportation industry in Croatia is outlined in the Transport Development Strategy for the period 2017–2030, which defines the concepts of sector strategies (Ministry of the Sea, Transport, and Infrastructure of the Republic of Croatia 2017). Traffic and mobility are included as one of the five priority thematic areas of the Croatian Smart Specialization Strategy. It confirms the importance of the transportation sector in the context of economic and social development. Transport directly affects the expansion of the industrial market, and indirectly affects economic growth. It improves living standards and competitiveness among regions and local communities, but also improves the physical expansion and integration of infrastructure (Ministry of Economy, Entrepreneurship and Crafts of the Republic of Croatia 2016). In order to prioritize the government-driven strategic development projects, there is a need for an economic impact analysis of transportation sectors, given the excessive costs of transport infrastructure, to provide policymakers with feasible and the most scientifically proven information on the economic impact of transportation industries (Lee and Yoo 2016).

The objective of this paper is to quantify multiplicative effects of the Croatian transportation sector and to identify changes and trends for the period 2004–2015 by using input–output (IO) analysis. The output and gross value added (GVA) multipliers for the Croatian economy are estimated. Comparative analyses with selected European Union (EU) countries are conducted to assess the Croatian transport industry position from an international perspective. The contribution of this paper is intended to address the lack of recent studies which quantify the economic effects of transport on the Croatian economy. The estimation of multiplicative effects is especially important when an impact of exogenous shock is the subject of the research. The current COVID-19 pandemic resulted in policy measures which have restricted the movements and contacts of humans. A recent study (Fernandes 2020) concluded that service-oriented economies will record the most pronounced effects caused by the outbreak of the virus. The same author also highlighted the spillover of the crisis from transport and other service industries to the rest of the economy and the spread of negative effects throughout the value-added chain. Fornaro and Wolf (2020) showed that the spread of the epidemic resulted first in a shock of demand reduction, followed by a reduction in supply and continued negative spiral effects. Simulations of different scenarios of the impact of exogenous shock on the demand for transport services and the reduction in the GVA for the total economy based on the estimates of the multiplicative effect are also provided in this research.

The paper layout is organized into five sections. After the Introduction, in Section 2, we provide an overview of recent literature which includes IO applications to explore transportation–economic linkages. In Section 3, we provide the methodology concept and modeling framework to quantify the economic effects of the transportation sector. In Section 4, we present the results of our empirical analysis, while in the last section the key conclusions are drawn along with recommendations for future applications.

*Importance of the Transportation Sector in Croatia*

The influence of the transport industry on the development and growth of national economies was the topic of numerous studies and recently published works by Tong and Yu (2018) and Jurgelane-Kaldava et al. (2019). The basis of sectoral development and an increase in quality and reliability of transport services are related to investments in transport infrastructure, which enable the prosperity and affirmation of the total transport system.

The Croatian transport sector uses various forms of transport, such as rail, road, water, and air, which have specific roles in providing support to passengers and providing freight services at the international and national level. Table 1 presents trends in annual freight traffic for six transportation subcategories in the period 2004–2015. Data show the dominant role of road transportation in total domestic freight transport with an average share of 58.7% in the observed period. It was declining after the economic crisis in 2009, and the indicators of mild recovery appeared at the end of the analyzed period. The negative trend is also evident in other transportation modes. High dependence on road transportation could lead to an increase in negative implications of transport on the environment as emissions of air pollutants or congestion (Bharadwaj et al. 2017), contributing to the overall negative externalities (Alises and Vassallo 2015). The Transport Development Strategy of the Republic of Croatia for the period 2014–2030 implemented policy measures to stimulate a shift to alternative transport modes and reduce negative impacts. It envisages diverting 30% by 2030, and by 2050 more than 50% of road freight transport over distances of more than 300 km to rail, sea, and inland waterways through the construction of green freight corridors (Government of the Republic of Croatia 2014).

**Table 1.** Annual freight traffic by transportation mode in the period 2004–2015 (in million tons and percentage of overall share).

| | 2004 | 2005 | 2006 | 2007 | 2008 | 2009 | 2010 | 2011 | 2012 | 2013 | 2014 | 2015 |
|---|---|---|---|---|---|---|---|---|---|---|---|---|
| **Railway transport** | 12.23 (11.2) | 13.91 (12.2) | 15.40 (12.9) | 15.73 (12.6) | 14.85 (9.0) | 11.79 (8.1) | 12.20 (9.5) | 11.95 (9.5) | 11.14 (10.1) | 10.87 (9.8) | 10.39 (10.0) | 9.94 (9.3) |
| **Road transport** | 55.32 (50.5) | 58.89 (51.9) | 63.84 (53.3) | 66.81 (53.4) | 110.81 (67.0) | 92.85 (63.8) | 74.97 (58.3) | 74.65 (59.6) | 65.44 (59.6) | 67.50 (60.6) | 66.15 (63.4) | 66.49 (62.4) |
| **Seawater and coastal transport** | 31.23 (28.5) | 29.98 (26.4) | 31.42 (26.2) | 32.42 (25.9) | 30.77 (18.6) | 31.37 (21.6) | 31.95 (24.8) | 30.35 (24.2) | 25.64 (23.4) | 24.74 (22.2) | 20.34 (19.5) | 21.38 (20.1) |
| **Inland waterway transport** | 0.90 (0.8) | 1.40 (1.2) | 0.40 (0.3) | 0.38 (0.3) | 0.27 (0.2) | 0.26 (0.2) | 0.52 (0.4) | 0.50 (0.4) | 0.65 (0.6) | 0.58 (0.5) | 0.49 (0.5) | 0.57 (0.5) |
| **Air transport** | 0.01 (0.0) | 0.01 (0.0) | 0.01 (0.0) | 0.01 (0.0) | 0.01 (0.0) | 0.00 (0.0) | 0.00 (0.0) | 0.00 (0.0) | 0.00 (0.0) | 0.00 (0.0) | 0.00 (0.0) | 0.00 (0.0) |
| **Transport via pipelines** | 9.88 (9.0) | 9.40 (8.3) | 8.64 (7.2) | 9.69 (7.7) | 8.77 (5.3) | 9.20 (6.3) | 8.94 (7.0) | 7.77 (6.2) | 6.88 (6.3) | 7.62 (6.8) | 6.92 (6.6) | 8.16 (7.7) |
| **Goods carried** | 109.56 | 113.57 | 119.71 | 125.04 | 165.47 | 145.47 | 128.57 | 125.22 | 109.74 | 111.31 | 104.28 | 106.54 |

Source: Data from CBS (2020a).

The average annual share of the transportation and storage sector in a total number of persons employed in legal entities in Croatia amounted to 6% during the period 2004–2015. Total freight volume, investments, and employment confirm the dominant role of road transportation. In 2015, 46.3% of total employees in legal entities of the transportation and storage sector worked in the subsector of land transport and transport via pipelines, 30.2% in warehousing and support activities for transportation, 17.5% to postal and courier services, and solely 4.5% employed in water transport, and respectively 1.6% in the air transport subsector (CBS 2020a).

Table 2 presents the comparison of the economic structure of selected old and new EU member states (NMS). Old member group includes Germany (DE), Italy (IT), Spain (ES), and the United Kingdom (UK), which has been one of the strongest EU economies before Brexit. The group of selected NMS economies includes economies that are similar to Croatia (HR) according to the population size and geographical location: Slovenia (SL), Slovakia (SK), Hungary (HU), and Czech Republic (CZ). The share of agriculture and industry in GVA is generally higher in the NMS group while more developed EU old members recorded a higher part of the public, business, and personal services. The hotel industry and trade significance are highest in Spain and Croatia due to geographical and climate conditions favoring tourism. The share of transport in total GVA of selected

economies varies from 4.0% in the United Kingdom to 6.4% in Slovenia. Transportation is a more significant economic sector in NMS economies, while its share in the old members is slightly below. Land transport and supporting services recorded a dominant share of GVA created in the transport sector in all analyzed economies.

**Table 2.** Economic structure of selected EU economies, in percentage of GVA in 2018.

|  | UK | DE | IT | ES | SK | HU | CZ | SL | HR |
|---|---|---|---|---|---|---|---|---|---|
| **Agriculture** | 0.6 | 0.7 | 2.2 | 3.1 | 2.7 | 4.1 | 2.1 | 2.6 | 3.6 |
| **Industry** | 13.9 | 25.5 | 19.7 | 16.1 | 24.7 | 24.8 | 29.7 | 26.7 | 19.6 |
| **Construction** | 6.4 | 4.9 | 4.2 | 6.1 | 7.9 | 5.1 | 5.6 | 5.7 | 5.4 |
| **Trade; Hotels** | 13.4 | 11.6 | 15.8 | 19.0 | 12.0 | 12.3 | 13.1 | 14.6 | 18.8 |
| **Public serv.** | 18.1 | 18.3 | 16.6 | 17.9 | 14.7 | 16.9 | 15.1 | 16.1 | 15.6 |
| **Busines and personal serv.** | 43.5 | 34.6 | 36.0 | 33.2 | 31.7 | 30.8 | 28.7 | 27.8 | 32.1 |
| **Transport** | 4.0 | 4.4 | 5.5 | 4.6 | 6.3 | 6.1 | 5.7 | 6.4 | 4.8 |
| of which |  |  |  |  |  |  |  |  |  |
| **Land** | 1.5 | 1.7 | 2.8 | 2.1 | 3.8 | 3.1 | 2.9 | 3.3 | 2.4 |
| **Water** | 0.4 | 0.2 | 0.2 | 0.1 | 0.0 | 0.0 | 0.0 | 0.1 | 0.4 |
| **Air** | 0.3 | 0.2 | 0.2 | 0.3 | 0.0 | 0.6 | 0.1 | 0.1 | 0.1 |
| **Supporting services** | 1.2 | 1.7 | 2.1 | 1.8 | 2.1 | 1.9 | 2.3 | 2.5 | 1.4 |
| **Postal serv.** | 0.6 | 0.5 | 0.2 | 0.2 | 0.4 | 0.5 | 0.4 | 0.5 | 0.5 |

Source: Data from Eurostat (2020a).

## 2. Literature Review

This section provides an overview of research on the economic impact of the transportation sector using IO analysis. The available literature presents some basic assumptions (Gretton 2013; Gupta 2009) and limitations (Miller and Blair 2009) of the IO model. Yu (2017) provided a comprehensive overview of IO model applications to economic linkages of transportation. Recently, Morrissey and O'Donoghue (2013) and Lee and Yoo (2016) have applied it in identifying the role of transport clusters. Some studies focused on specified transport subsectors. Wang and Wang (2019) and Santos et al. (2018) explored the port industry significance. The economic effects of the cruise industry were examined by Vayá et al. (2017) and Chang et al. (2015). Oxford Economics (2017) researched the Croatian shipping industry role that is not directly classified to transport, but for which performance is strongly related to water transport services. Bagoulla and Guillotreau (2020) analyzed the impact of maritime transport in France on the domestic economy, providing a different perspective by assessing the environmental impact of shipping on direct and indirect gas emissions. Yu et al. (2019) constructed the China non-competitive constant price IO model comprising the transport and storage sector. Kwak et al. (2005) examined the status and economic impact of four maritime industries in Korea to present policymakers' relations of these sectoral industries with the rest of the national economy. The Portuguese maritime cluster was assessed using three different qualitative and quantitative methodologies (Salvador et al. 2016), including IO analysis, indicating intra-sectoral relations of the marine industry as significant while emphasizing weak intermediate linkages. The interaction between air transport and economic development in Greece was studied by Dimitrios et al. (2017) and Dimitrios and Maria (2018). Studies quantified the socio-economic impact and the level of dependence of regions heavily relying on tourism. The results showed the importance of air transport for the Greek economy, mainly due to the high dependency and correlation between tourism and air connectivity, creating a high indirect effect on the national economic model. Stebbings et al. (2020) used IO data in quantifying the contribution of the marine sector to the United Kingdom economy. The results reveal

the twice as much estimated contribution of the marine economy to the overall United Kingdom economy if indirect effects are included. The economic impact of the marine sector of Australian coastal communities was an objective of van Putten et al. (2016). By the IO model, authors identified the interrelationships of the different maritime sector activities by highlighting key industries that could retain the current level or, perhaps, secure the marine sector's future growth. Chiu and Lin (2012) explored the maritime industry effects within other parts of the Taiwan economy. The study reveals the questionable position of industry in the domestic economy, considering the economic impact and the low intensity of dispersion.

The economic effects of final demand on production, GVA, and employment have been estimated for individual sectors of the Croatian economy. Buturac et al. (2017) found the highest output multipliers in Croatia for the construction (1.68) and manufacturing industries (1.599). Multiplicative effects for agriculture, estimated to be 1.54, are close to the national average (1.53). The lowest multipliers have been found for public and personal services, which are labor-intensive low-tech industries. Keček et al. (2019) found that ICT sectors contributed to the total Croatian GVA at a range higher than 4.5% if indirect effects are included. Ivandić and Šutalo (2019) estimated the GVA multiplier for Croatian tourism at 1.55 and the total contribution of tourism to 16.9% in 2016. In Mikulić et al. (2018), wind-power plant deployment effects were quantified, where small multiplicative ones have been found due to the high import content of high-tech products required in those plants. Mikulić et al. (2020) valorized economic effects from the energy renovation program of public buildings in Croatia and by application of closed IO model estimated investment multiplier at 2.5. As the analysis of the Croatian transportation sector's significance has not yet been adequately evaluated, there is a need to provide a quantitative analysis of its total effects on the Croatian economy.

## 3. Methodology

### 3.1. General Structure of Input–Output Analysis

The relation between transportation and other economic sectors and the economic impact of the transport industry on the national economy has been examined by various analytical frameworks developed for specific purposes. Cost-benefit analysis (CBA) and IO analysis are the most frequently used approaches (Yu 2017). While the central focus of the CBA is on direct access within the contribution to the transport sector, IO analysis applies to the structure of more extensive national economic impacts and linkages between specific activities (Lakshmanan 2011). IO analysis enables quantitative macroeconomic insight to assess the influence of final demand on domestic production, GVA, and employment. Leontief (1986) was the first who developed this approach, proposing the inter-sectoral model that provides linkages among productive industries of a specified economy on a national or regional level. It has been widely used in analyzing the impact of multiple areas in recent years (Miller and Blair 2009). Using IO tables as a statistical foundation, mathematical relations of inter-industry transaction tables were created by applying the Leontief inverse matrix. It should note that the structure of IO tables of a specified economy is divided into several productive sectors, where columns represent input values of particular sectors and rows represent respective output values. The impact of cross-sector flows on the overall production of each sector is determined in the IO table by the principal equation of the IO model (Miller and Blair 2009):

$$x_i = \sum z_{ij} + f_i \qquad (1)$$

where $x_i$ is a total output of sector $i$, $z_{ij}$ represents the number of a product from sector $i$ used as an intermediate input in production by sector $j$, and $f_i$ represents a final demand of sector $i$, for $i, j = 1, \ldots, n$ ($n$ is a number of sectors). This equation represents a system of linear equations, one per sector of the economy, where the output of each sector is divided between intermediate products and final demand. The relation between inputs used by

sectors and the total produced output is determined with technical coefficients $a_{ij} = \frac{z_{ij}}{x_j}$. By using simple matrix notation, the system of Equation (1) for the total economy, it is possible to rewrite it as

$$x = Ax + f \tag{2}$$

where $A$ is $n \times n$ matrix of technical coefficients, $x$ is the column vector of outputs, and $f$ is the column vector of final demands, i.e.,

$$x = \begin{bmatrix} x_1 \\ \vdots \\ x_n \end{bmatrix}, \quad A = \begin{bmatrix} a_{11} & \cdots & a_{1n} \\ \vdots & \ddots & \vdots \\ a_{n1} & \cdots & a_{nn} \end{bmatrix}, \quad \text{and } f = \begin{bmatrix} f_1 \\ \vdots \\ f_n \end{bmatrix} \tag{3}$$

The Equation (2) can be rewritten as

$$(I - A)x = f \tag{4}$$

where $I$ is the identity matrix, and $(I - A)$ and is called the Leontief matrix. The solution to this system of linear equations is:

$$x = (I - A)^{-1}f \tag{5}$$

where $L = (I - A)^{-1}$ represents the Leontief inverse matrix or multiplier matrix. This matrix can be interpreted as the relation of direct and indirect requirements for the output of each sector to support one unit of deliveries to the final demand, and it is defined by elements $a_{ij}$.

The primary objective of this research is a calculation of the output and GVA multiplier for the transportation sector. In this research, an open IO model based on domestic demand is used. Grady and Muller (1988) argue the preference for using the open IO model instead of the closed one, which includes induced household expenditures. Multipliers are calculated as the relation of total (direct plus indirect) effects to direct effects. The simple output multiplier for the sector can be calculated by using the following relation:

$$m(o)_j = \sum_{i=1}^{n} l_{ij} \tag{6}$$

i.e., the output multiplier is calculated as a sum of individual industry column elements of the Leontief inverse matrix. Value-added multipliers measure the value-added of a single sector as a result of an additional output delivered to final demand. In matrix form, it can be denoted as

$$m(v) = v'_c L \tag{7}$$

where $v'_c$ is vector of value-added coefficients representing the share of GVA of each sector in its output.

### 3.2. Data Sources

The publication of official symmetric IO tables for Croatia facilitated the use and quantification of data for analytical purposes to assess the overall contribution of transportation in Croatia in the observed period. Data used for this research comprises four IO tables from different sources as follows:

- IO tables for 2004, based on the level of 60 aggregate sectors (CPA 2002 classification).
- IO tables for 2010, 2013, and 2015 based on the level of 64 aggregate sectors (CPA 2008 classification).

IO tables for 2004 and 2010 were retrieved from the Croatian Bureau of Statistics, while updated IO tables for the year 2013 were taken from Mikulić (2018). The IO table for 2015 is available from the Eurostat database (Eurostat 2020b). While correspondence of the old

and new classification systems is full for land, water, and air transport (Table 3), sector 64 in CPA 2002 is not fully comparable to H53, because telecommunication services are now classified into the new information and communication sector.

**Table 3.** Transport classified by modes in CPA 2002 and CPA 2008.

| CPA 2002 | CPA 2008 |
| --- | --- |
| 60 Land transport services and transport services via pipelines | CPA_H49 Land transport services and transport services via pipelines |
| 61 Water transport services | CPA_H50 Water transport services |
| 62 Air transport services | CPA_H51 Air transport services |
| 63 Supporting and auxiliary transport services; travel agency services | CPA_H52 Warehousing and support services for transportation |
| 64 Postal and telecommunication services | CPA_H53 Postal and courier services |

Source: Data from Eurostat (2020c).

In this research, comparative analyses of the transportation sector are provided for the selected new EU members (Hungary, Czech Republic, Slovakia, and Slovenia) and the selected old EU member states (Germany, Italy, United Kingdom, and Spain). IO tables for those countries are retrieved from the Eurostat database (Eurostat 2020b). The last available IO data refer to 2015, which could affect the main assumptions of the IO method on the existence of fixed technological coefficients in a more recent period. The technology could be changed consequently to the implementation of more efficient production processes, the use of modern ICT technologies, changes of relative prices, and other factors (Miller and Blair 2009). The dynamic I-O analysis with coefficients updated by application of statistical techniques, such as the RAS method or Cross-Entropy Model, has been described in the economic literature (Miller and Blair 2009). However, if only statistical methods are applied, without the inclusion of more recent official data on the change of the structure of intermediate consumption, it could harm the reliability of estimates. Rokicki et al. (2020) found noticeable differences at the sectoral level comparing survey-based versus algorithm-based multi-regional IO tables. As a set of EU economies is included in the sample, dynamization of IO data based exclusively on statistical techniques could result in estimates which are not robust and depend on the statistical technique arbitrarily selected by the authors.

## 4. Research Results

Indirect effects of an economic sector in the IO model result from the technical requirements of a production process applied and the structure of domestic and imported intermediate inputs used. The higher share of domestic inputs incorporated in the sector's output implies higher integration of the domestic economy and larger indirect effects. This section presents analyses of multiplicative effect trends of the Croatian transportation sector and a comparison to the selected set of EU economies.

### 4.1. The Structure of Output in the Transport Sector in Croatia and EU

The most significant input required by companies operating in the transport sector is energy. In most EU economies, energy balances reveal their dependence on imported energy, especially crude oil and oil derivatives. As a result of import dependence, a specified share of indirect effects is not operating in the domestic economy, but is transferred abroad. A comparison of the output structures in the transportation sectors in Croatia and selected EU countries is provided by Table A1 in the Appendix A. Figure 1 represents the comparison of land transport as the most significant subsector.

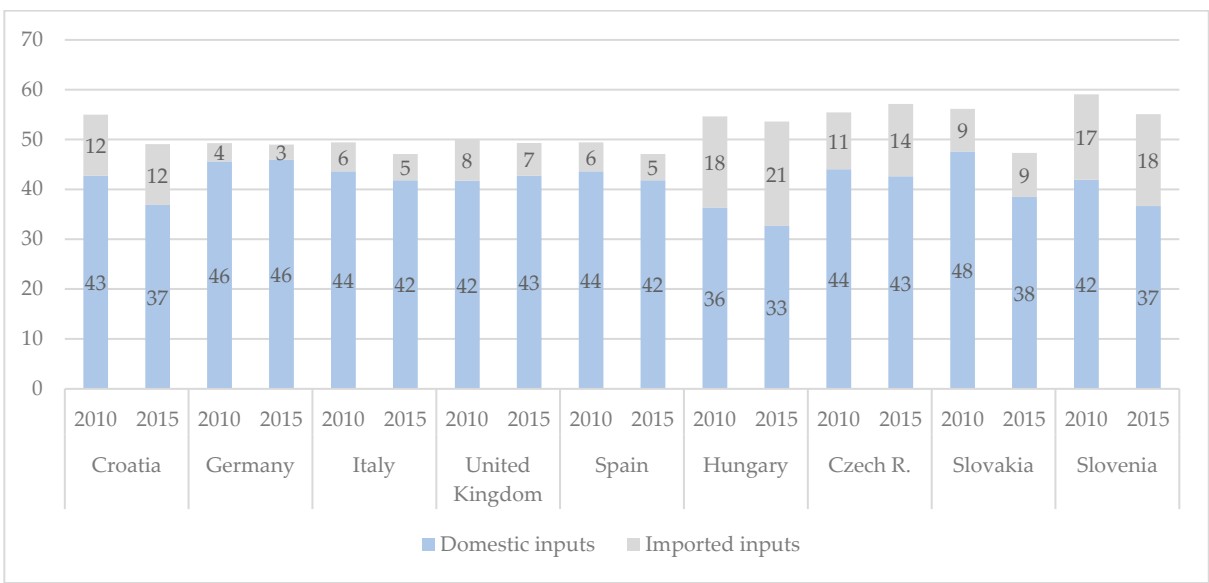

**Figure 1.** Domestic and imported inputs used in the land transport services and transport services via pipelines, in% of gross output. Source: Authors' calculation.

A general declining trend of the share of domestic intermediate consumption can be noticed in both land and water transport sectors in the analyzed period in most of the economies (Figure 1). The structure of inputs used by Croatian land transport companies is more similar to new EU member states, where the share of imports is higher compared to more developed old EU countries.

Warehousing and supporting transport services in most EU economies are more integrated with domestic producers. The share of imported inputs in this sector is the highest in Hungary, the Czech Republic, and Croatia. Land transport services and transport services via pipelines and air transport sector (presented in Table A1 in Appendix A) recorded an average level share among analyzed countries. Otherwise, water transport, warehousing, and support services for transportation and postal and courier services were among the lowest, which can be explained with the relatively reduced integrity level of the Croatian economy into the overall European market.

*4.2. Output and GVA Multipliers of Croatian Transportation Sector in Period 2004–2015*

Multiplier analysis of the transportation sector is performed based on the IO tables for 2004, 2010, 2013, and 2015. As the transportation industry for its operation uses intermediate inputs delivered by other activities, it induces spillover effects on the total economy. Table 4 shows output multipliers for the Croatian transportation sector for 2004, 2010, 2013, and 2015.

**Table 4.** Output multipliers for the Croatian transportation sector.

| Sector Code | | **2004** | **2010** | **2013** | **2015** |
|---|---|---|---|---|---|
| 60 | CPA_H49 | 1.69 | 1.66 | 1.67 | 1.58 |
| 61 | CPA_H50 | 1.73 | 1.60 | 1.79 | 1.55 |
| 62 | CPA_H51 | 1.91 | 1.82 | 2.09 | 1.85 |
| 63 | CPA_H52 | 1.71 | 1.55 | 1.69 | 1.62 |
| 64 | CPA_H53 | 1.54 | 1.31 | 1.35 | 1.45 |

Source: Authors' calculation.

In the observed period, the highest output multiplier was recorded for the air transport sector. In 2015, it amounted to 1.85. It means that if the final demand for products in this

sector increases by 1 HRK, the total output in the Croatian economy will grow by 1.85 HRK. Output multiplier values for land transport services and transport services via pipelines were the most consistent, while the lowest value was observed for the postal and courier services sector. Similarly, GVA multipliers for the transportation sector were calculated, considering four analyzed years, as shown in Table 5.

**Table 5.** GVA multipliers for the Croatian transportation sector (total effects/direct effects).

| | Sector Code | 2004 | 2010 | 2013 | 2015 |
|---|---|---|---|---|---|
| 60 | CPA_H49 | 1.61 | 1.70 | 1.68 | 1.59 |
| 61 | CPA_H50 | 1.67 | 1.67 | 1.78 | 1.56 |
| 62 | CPA_H51 | 2.29 | 2.24 | 3.15 | 2.63 |
| 63 | CPA_H52 | 1.69 | 1.49 | 1.60 | 1.64 |
| 64 | CPA_H53 | 1.49 | 1.19 | 1.22 | 1.30 |

Source: Authors' calculation.

GVA multipliers were mostly found at the same sectoral importance level as those indicated to output multipliers. The air transport sector had the highest values, and the postal and courier services sector had the lowest GVA multipliers.

*4.3. Comparison of Multiplicative Effects in Croatia and Selected EU Countries*

The competitiveness of the Croatian transportation sector could be assessed by the comparative analysis of output and GVA multipliers. The study includes the selected advanced EU economies and developing new member states. The analysis of output and GVA multipliers is based on the years 2010 and 2015, where the same classification of activities has been applied.

In most observed economies, the highest output multiplier is found in air transport and supporting transport services (Table 6). On the other hand, sector postal and courier services have the lowest output multipliers values. Output multipliers of the Croatian transportation sector were found among the lowest compared to selected old and new member states in the observed period. Inland transport, supporting transport, and postal services in the old EU members group generally recorded higher multipliers values. It can be explained, not only with higher integration of domestic producers, but also with country size and the dominant role of domestic over international transport. Transport companies in small economies, such as Croatian or Slovenian, usually participate in international transport operations with a higher proportion and buy a significant share of oil products and other intermediate inputs abroad. On the contrary, the lowest output multipliers were calculated for Hungary, Slovenia, and Croatia, while the largest fluctuations in output multiplier values were found in the water transport sector.

The total GVA results for the transportation sector in selected EU economies are presented in Table 7. Cumulative effects should be interpreted as GVA, which is created in the overall economy when final consumption for a specified transport sector increases by a unit monetary value. Total effects in most economies are highest for post services where 1 EUR increase of final demand results in 0.835 EUR GVA in the overall economy (2015 data). An increase of final demand for water and air transport results in lower amounts of domestic GVA, which can be explained by a higher share of international transport where a certain proportion of energy products and other intermediates are bought abroad.

**Table 6.** Output multipliers for the transportation sector in selected European economies.

| | CPA_H49 | | CPA_H50 | | CPA_H51 | | CPA_H52 | | CPA_H53 | |
| --- | --- | --- | --- | --- | --- | --- | --- | --- | --- | --- |
| | **2010** | **2015** | **2010** | **2015** | **2010** | **2015** | **2010** | **2015** | **2010** | **2015** |
| UNITED KINGDOM | 1.69 | 1.71 | 1.95 | 1.87 | 1.60 | 1.51 | 1.90 | 1.92 | 1.47 | 1.56 |
| GERMANY | 1.84 | 1.82 | 1.31 | 1.58 | 1.87 | 1.56 | 1.96 | 1.92 | 1.73 | 1.93 |
| ITALY | 1.77 | 1.87 | 2.22 | 2.02 | 2.02 | 2.42 | 1.94 | 1.97 | 1.73 | 1.70 |
| SPAIN | 1.79 | 1.74 | 1.95 | 2.05 | 1.95 | 2.05 | 1.98 | 1.90 | 1.72 | 1.78 |
| SLOVAKIA | 1.86 | 1.63 | 1.75 | 1.70 | 1.51 | 1.74 | 2.54 | 1.79 | 1.73 | 1.89 |
| HUNGARY | 1.53 | 1.45 | 1.42 | 1.33 | 1.24 | 1.10 | 1.48 | 1.42 | 1.30 | 1.26 |
| CZECH REPUBLIC | 1.77 | 1.73 | 2.19 | 2.02 | 2.06 | 1.98 | 1.97 | 1.94 | 1.58 | 1.63 |
| SLOVENIA | 1.69 | 1.58 | 1.33 | 1.10 | 1.87 | 1.88 | 1.77 | 1.81 | 1.28 | 1.44 |
| CROATIA | 1.66 | 1.58 | 1.60 | 1.55 | 1.82 | 1.85 | 1.55 | 1.62 | 1.31 | 1.45 |
| **AVERAGE** | 1.73 | 1.68 | 1.75 | 1.69 | 1.77 | 1.79 | 1.90 | 1.81 | 1.54 | 1.63 |

Source: Authors' calculation.

**Table 7.** Total gross value added (GVA) effects for the transportation sector in selected European economies.

| | CPA_H49 | | CPA_H50 | | CPA_H51 | | CPA_H52 | | CPA_H53 | |
| --- | --- | --- | --- | --- | --- | --- | --- | --- | --- | --- |
| | **2010** | **2015** | **2010** | **2015** | **2010** | **2015** | **2010** | **2015** | **2010** | **2015** |
| UNITED KINGDOM | 0.813 | 0.828 | 0.740 | 0.787 | 0.657 | 0.689 | 0.860 | 0.867 | 0.818 | 0.791 |
| GERMANY | 0.859 | 0.857 | 0.451 | 0.466 | 0.585 | 0.546 | 0.801 | 0.777 | 0.806 | 0.808 |
| ITALY | 0.802 | 0.841 | 0.724 | 0.725 | 0.564 | 0.642 | 0.828 | 0.848 | 0.865 | 0.844 |
| SPAIN | 0.822 | 0.825 | 0.748 | 0.824 | 0.614 | 0.694 | 0.874 | 0.874 | 0.926 | 0.919 |
| SLOVAKIA | 0.746 | 0.746 | 0.721 | 0.753 | 0.612 | 0.742 | 0.777 | 0.806 | 0.887 | 0.835 |
| HUNGARY | 0.670 | 0.655 | 0.381 | 0.373 | 0.239 | 0.461 | 0.792 | 0.720 | 0.841 | 0.822 |
| CZECH REPUBLIC | 0.714 | 0.653 | 0.687 | 0.552 | 0.582 | 0.472 | 0.787 | 0.706 | 0.807 | 0.755 |
| SLOVENIA | 0.641 | 0.626 | 0.502 | 0.312 | 0.514 | 0.538 | 0.871 | 0.815 | 0.926 | 0.869 |
| CROATIA | 0.720 | 0.717 | 0.604 | 0.697 | 0.696 | 0.619 | 0.770 | 0.726 | 0.884 | 0.871 |
| **AVERAGE** | 0.754 | 0.750 | 0.618 | 0.610 | 0.562 | 0.601 | 0.818 | 0.793 | 0.862 | 0.835 |

Source: Authors' calculation.

GVA multipliers, which are to be interpreted as a ratio of total effects created in the national economy and direct effects recorded in the transport sector, are similar to output multipliers, but with higher fluctuations in values, especially for the water transport and air transport sectors (Table 8). In air transport, the highest GVA multipliers are mostly dependent on low direct effects and the low margin charged by air transporters because of the extremely competitive market, while indirect effects are relatively high.

Transportation requires a significant input of imported products, primarily oil derivatives, used in the transport vehicles operations. Table 9 presents the total requirements for imported products per unit value of the transport industry output. The highest import requirements are estimated for air and water transport, and the lowest for postal services. Import requirements of Croatian transporters are similar to the average ones estimated for other EU economies. A decrease in import content of Croatian water transport in 2015 can be explained by the restructuring of this activity from international to local transport (ferries and touristic routes) due to the increased demand of foreign tourists visiting Croatia.

**Table 8.** GVA multipliers for the transportation sector in selected European economies (total effects/direct effects).

| | CPA_H49 | | CPA_H50 | | CPA_H51 | | CPA_H52 | | CPA_H53 | |
|---|---|---|---|---|---|---|---|---|---|---|
| | **2010** | **2015** | **2010** | **2015** | **2010** | **2015** | **2010** | **2015** | **2010** | **2015** |
| UNITED KINGDOM | 1.71 | 1.72 | 2.44 | 2.16 | 1.83 | 1.62 | 2.01 | 2.06 | 1.42 | 1.56 |
| GERMANY | 1.77 | 1.75 | 1.41 | 2.19 | 2.23 | 1.79 | 2.23 | 2.23 | 1.74 | 2.14 |
| ITALY | 1.66 | 1.76 | 2.86 | 2.27 | 2.86 | 8.35 | 1.94 | 1.99 | 1.57 | 1.58 |
| SPAIN | 1.79 | 1.73 | 2.42 | 2.67 | 2.73 | 3.15 | 2.10 | 1.99 | 1.69 | 1.79 |
| SLOVAKIA | 1.84 | 1.57 | 1.48 | 1.70 | 1.50 | 1.81 | 3.32 | 1.80 | 1.64 | 1.90 |
| HUNGARY | 1.60 | 1.48 | 2.04 | 1.83 | 1.76 | 1.10 | 1.40 | 1.40 | 1.21 | 1.19 |
| CZECH REPUBLIC | 1.74 | 1.73 | 3.06 | 3.13 | 3.66 | 5.17 | 1.95 | 2.06 | 1.51 | 1.59 |
| SLOVENIA | 1.97 | 1.76 | 1.39 | 1.19 | 8.36 | 6.25 | 1.69 | 1.85 | 1.21 | 1.38 |
| CROATIA | 1.70 | 1.59 | 1.67 | 1.56 | 2.24 | 2.63 | 1.49 | 1.64 | 1.19 | 1.30 |
| **AVERAGE** | 1.75 | 1.68 | 2.09 | 2.08 | 3.02 | 3.54 | 2.01 | 1.89 | 1.46 | 1.60 |

Source: Authors' calculation.

**Table 9.** Total requirements of imported products for operations in the transportation sector, in selected European economies, per unit value of output.

| | CPA_H49 | | CPA_H50 | | CPA_H51 | | CPA_H52 | | CPA_H53 | |
|---|---|---|---|---|---|---|---|---|---|---|
| | **2010** | **2015** | **2010** | **2015** | **2010** | **2015** | **2010** | **2015** | **2010** | **2015** |
| UNITED KINGDOM | 0.15 | 0.14 | 0.24 | 0.19 | 0.28 | 0.24 | 0.13 | 0.12 | 0.14 | 0.15 |
| GERMANY | 0.11 | 0.11 | 0.54 | 0.53 | 0.41 | 0.44 | 0.16 | 0.19 | 0.15 | 0.15 |
| ITALY | 0.15 | 0.13 | 0.25 | 0.24 | 0.42 | 0.34 | 0.15 | 0.14 | 0.09 | 0.11 |
| SPAIN | 0.12 | 0.12 | 0.24 | 0.17 | 0.38 | 0.29 | 0.11 | 0.11 | 0.01 | 0.06 |
| SLOVAKIA | 0.21 | 0.19 | 0.20 | 0.20 | 0.32 | 0.23 | 0.21 | 0.15 | 0.11 | 0.15 |
| HUNGARY | 0.28 | 0.31 | 0.56 | 0.58 | 0.75 | 0.54 | 0.18 | 0.23 | 0.11 | 0.12 |
| CZECH REPUBLIC | 0.24 | 0.29 | 0.26 | 0.39 | 0.33 | 0.43 | 0.18 | 0.26 | 0.14 | 0.18 |
| SLOVENIA | 0.26 | 0.26 | 0.49 | 0.68 | 0.40 | 0.35 | 0.15 | 0.16 | 0.06 | 0.09 |
| CROATIA | 0.23 | 0.21 | 0.36 | 0.21 | 0.25 | 0.24 | 0.20 | 0.21 | 0.10 | 0.10 |
| **AVERAGE** | 0.19 | 0.20 | 0.35 | 0.35 | 0.39 | 0.34 | 0.16 | 0.17 | 0.10 | 0.12 |

Source: Authors' calculation.

*4.4. Simulation of Total GVA Effects Caused by a Reduction of Transport Services Due to Restrictions in Movements of Persons as a Result of COVID-19 Pandemic*

Multipliers are estimated by the IO method are efficient in both directions. Transport activity reduction indirectly affects other domestic companies included in the value-added chain of the transport industry. Although data for all transport modes are not regularly published for all economies, it is clear that the reduction of the volume of transport activity in 2020 will be significant. Available data on passenger–kilometers realized in rail transport in the second quarter of 2020 in Croatia and Germany dropped to only one third compared to the same period of 2019. Even worse performance of rail transport is recorded in Spain, France, and Italy, where the reduction amounted to over 80%. According to Air Passenger Market Analyses (IATA 2020), passenger air transport measured in revenue passenger kilometers was down 90% year-on-year in April 2020, and 75% in August. Although the transport of freight recorded a modest reduction, GVA data in transport activity will certainly point to a severe reduction when available next year. Table 10 presents the simulation results on the total national GVA reduction due to transport activities reduction caused by the COVID-19 pandemic. According to all three scenarios, the worst effects are

expected for Slovenia and Italy. According to the moderate scenario of transport activity reduction of 35%, the result will be a decrease in total economic activity in the range from 2.6 to 4.1%, when indirect effects are included. As the European Commission (2020b) in Autumn Forecast estimates the average growth rate of economic activity in EU at 7.4%, it is clear that one third to one half of the reduction of GDP could be related to the poor performance of the transport industry under the impact of exogenous shock.

**Table 10.** Simulation of the effects of the total national GVA reduction in 2020 due to the decreasing of transport activity.

| Reduction in Transport Activity | −20% | −35% | −50% |
|---|---|---|---|
| | **Reduction in Total National GVA** | | |
| UNITED KINGDOM | −1.5 | −2.6 | −3.7 |
| GERMANY | −1.8 | −3.1 | −4.4 |
| ITALY | −2.2 | −3.9 | −5.6 |
| SPAIN | −1.8 | −3.1 | −4.5 |
| SLOVAKIA | −2.1 | −3.7 | −5.3 |
| HUNGARY | −1.7 | −3.0 | −4.2 |
| CZECH REPUBLIC | −2.1 | −3.8 | −5.4 |
| SLOVENIA | −2.3 | −4.1 | −5.8 |
| CROATIA | −1.5 | −2.7 | −3.9 |

Source: Authors' calculation.

## 5. Discussion and Conclusions

The significance of transportation for the Croatian economic growth was examined by IO analysis. It was used to determine the integration of transportation and other domestic sectors. The utilization of modern and efficient transportation leads to significant influence on the growth of other economic activities and socio-economic development of Croatia. Multiplicative effects in the transportation sector are notable in the observed period, especially for the air transport sector. While output multipliers for road and water transport are close to average multipliers for all economic sectors found in recent literature (Buturac et al. 2017), the multiplier for air transport is significantly higher due to more complex applied technology. The lowest multipliers for Croatia and sampled economies have been detected for postal and courier services, relatively simple labor-intensive activities. The moderate intensity of output multipliers was in water transport, land transport services, and transport services via pipelines and warehousing and support services. Postal and courier services recorded lower output multipliers. The highest GVA multipliers were recorded for air transport services, while the lowest ones were recorded for postal and courier services.

The effects of the transportation sector analyzed in this research are primarily distributed through other activities rather than within the transportation cluster, meaning that indirect effects prevail as opposed to direct ones, spreading across various activities, especially for air and water transport sectors. The transportation sector, identified as a loose network of interrelated activities, shows a relatively moderate degree of integration in the whole economy. In perspective, a higher level of integrity and connections with other industries is needed, on a national and international level, which would generate higher value-added and other multiplicative effects along with the influence on the achievement of broader socio-economic goals. The international market trends, indicated in 2010, have been marked by declining demand and growing competition but, the recovery was perceivable concerning the multipliers increase in 2015.

Compared to other European countries, the Croatian transportation sector recorded lower output and GVA multipliers, which implies that other countries, like Italy, the United

Kingdom, Spain, or the Czech Republic, capitalized on the transportation sector more for growth and development than other countries. The Croatian transportation recorded a lower share of the imported intermediate and average level of the domestic inputs; a higher level of value-added, compared to the other examined European economies, is very similar to the Slovenian and Hungarian transportation industries.

Multipliers estimated in this study are beneficial not only in the positive direction connected to growth in final demand, but also in a sudden decrease due to exogenous shock. Simulation of the effects of the COVID-19 pandemic points to the transport industry as one of the principal sectors which caused a sharp decline of economic activity in EU economies.

More investment in the technological modernization of transportation to increase the competitiveness and share of higher value-added services is necessary. Utilizing the more sophisticated transportation level leads to the higher multiplicative effect of this sector and enable valorization of complete high-value base and human resources quality.

IO analysis of the transportation sector has proved very useful, and results were in line with the expectations. The unavailability of more recent data to perform longer-term IO analysis and the calculation of remaining multipliers associated with IO tables are the main limitations of this research. The recommendations for future research are, mainly, directed to the inclusion of alternative economic IO approaches and modeling applications to determine transportation–economic linkages, which would enable more detailed insight and perspective in the long-term.

**Author Contributions:** Conceptualization, L.V. and D.K.; methodology, L.V.; validation, D.M.; formal analysis, D.K.; investigation, L.V., D.K., and D.M.; resources, D.K.; data curation, D.K.; writing—original draft preparation, L.V.; writing—review and editing, D.K. and D.M.; visualization, D.K.; supervision, D.M. All authors have read and agreed to the published version of the manuscript.

**Funding:** This research received no external funding.

**Institutional Review Board Statement:** Not applicable.

**Informed Consent Statement:** Not applicable.

**Data Availability Statement:** The data presented in this study are available on request from the corresponding author.

**Conflicts of Interest:** The authors declare no conflict of interest.

## Appendix A

**Table A1.** The share of domestic and import inputs in the output of the transport sector.

| | 2010 | | | | | 2015 | | | | |
|---|---|---|---|---|---|---|---|---|---|---|
| | **Land** | **Water** | **Air** | **Warehousing** | **Postal Services** | **Land** | **Water** | **Air** | **Warehousing** | **Postal Services** |
| Croatia | | | | | | | | | | |
| Domestic inputs | 42.7 | 38.6 | 52.7 | 35.3 | 19.4 | 36.9 | 35.2 | 53.8 | 38.9 | 27.4 |
| Imported inputs | 12.3 | 23.2 | 13.0 | 11.6 | 5.6 | 12.2 | 12.3 | 11.2 | 12.3 | 3.8 |
| GVA | 42.3 | 36.2 | 31.0 | 51.8 | 74.6 | 45.0 | 44.7 | 23.5 | 44.4 | 67.1 |
| Germany | | | | | | | | | | |
| Domestic inputs | 45.6 | 17.6 | 45.6 | 53.3 | 41.0 | 46.0 | 32.6 | 32.5 | 52.0 | 52.9 |
| Imported inputs | 3.7 | 50.3 | 28.2 | 8.3 | 9.4 | 3.0 | 46.1 | 36.1 | 10.8 | 6.9 |
| GVA | 48.5 | 32.1 | 26.2 | 36.0 | 46.4 | 48.9 | 21.3 | 30.6 | 34.9 | 37.7 |

**Table A1.** *Cont.*

| | | 2010 | | | | | 2015 | | | |
|---|---|---|---|---|---|---|---|---|---|---|
| | **Land** | **Water** | **Air** | **Warehousing** | **Postal Services** | **Land** | **Water** | **Air** | **Warehousing** | **Postal Services** |
| | | | | | Italy | | | | | |
| Domestic inputs | 43.6 | 56.9 | 55.3 | 53.5 | 42.4 | 41.8 | 61.8 | 62.0 | 50.7 | 45.3 |
| Imported inputs | 5.8 | 12.1 | 22.1 | 4.3 | 1.7 | 5.2 | 7.1 | 15.4 | 4.7 | 1.8 |
| GVA | 45.9 | 30.9 | 22.5 | 41.7 | 54.7 | 47.7 | 30.9 | 22.1 | 44.0 | 51.4 |
| | | | | | United Kingdom | | | | | |
| Domestic inputs | 41.7 | 53.7 | 36.6 | 51.4 | 29.2 | 42.8 | 51.2 | 32.2 | 52.4 | 35.0 |
| Imported inputs | 8.4 | 14.6 | 22.3 | 5.5 | 9.1 | 6.6 | 10.9 | 19.2 | 5.2 | 8.9 |
| GVA | 47.5 | 30.3 | 35.9 | 42.7 | 57.8 | 48.2 | 36.5 | 42.5 | 42.2 | 50.6 |
| | | | | | Spain | | | | | |
| Domestic inputs | 43.6 | 56.9 | 55.3 | 53.5 | 42.4 | 41.8 | 61.8 | 62.0 | 50.7 | 45.3 |
| Imported inputs | 5.8 | 12.1 | 22.1 | 4.3 | 1.7 | 5.2 | 7.1 | 15.4 | 4.7 | 1.8 |
| GVA | 45.9 | 30.9 | 22.5 | 41.7 | 54.7 | 47.7 | 30.9 | 22.1 | 44.0 | 51.4 |
| | | | | | Hungary | | | | | |
| Domestic inputs | 36.4 | 28.4 | 17.0 | 32.4 | 20.5 | 32.7 | 24.4 | 7.5 | 30.1 | 19.0 |
| Imported inputs | 18.3 | 48.0 | 69.0 | 9.1 | 5.7 | 20.9 | 50.8 | 50.4 | 14.7 | 7.1 |
| GVA | 41.8 | 18.7 | 13.6 | 56.6 | 69.3 | 44.2 | 20.4 | 41.9 | 51.4 | 68.8 |
| | | | | | Czech Republic | | | | | |
| Domestic inputs | 44.0 | 62.9 | 59.8 | 51.3 | 34.8 | 42.6 | 56.0 | 56.5 | 51.6 | 37.6 |
| Imported inputs | 11.4 | 10.8 | 16.5 | 6.5 | 7.9 | 14.5 | 21.8 | 26.5 | 12.5 | 9.6 |
| GVA | 41.0 | 22.4 | 15.9 | 40.4 | 53.4 | 37.8 | 17.6 | 9.1 | 34.2 | 47.6 |
| | | | | | Slovakia | | | | | |
| Domestic inputs | 47.6 | 36.4 | 33.6 | 71.0 | 41.8 | 38.5 | 41.9 | 44.2 | 47.0 | 50.3 |
| Imported inputs | 8.6 | 8.4 | 19.4 | 5.9 | 4.1 | 8.8 | 10.5 | 13.5 | 6.2 | 5.4 |
| GVA | 40.6 | 48.6 | 40.7 | 23.4 | 54.0 | 47.4 | 44.3 | 41.0 | 44.8 | 43.8 |
| | | | | | Slovenia | | | | | |
| Domestic inputs | 41.9 | 23.7 | 53.4 | 47.0 | 19.2 | 36.7 | 6.2 | 55.8 | 48.2 | 29.1 |
| Imported inputs | 17.1 | 39.6 | 31.4 | 5.4 | 3.4 | 18.4 | 66.7 | 25.8 | 7.0 | 4.6 |
| GVA | 32.5 | 36.2 | 6.1 | 51.6 | 76.8 | 35.6 | 26.3 | 8.6 | 44.2 | 63.0 |

Source: Authors' calculation.

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
