# Peer review of "The Impact of Transportation on the Croatian Economy: The Input–Output Approach"

_economies, doi:10.3390/economies9010007_

Round 1
Reviewer 1 Report
The article is ready to publish once some minor changes are made.
We judge mandatory to change notation in order to make it compatible with Miller and Blair (2009) and to specify that calculations are made on the basis of the domestic demand model.
It would be better if formula n. 7 was expressed in matrix notation.
I suggest the elimination of the analysis concerning the employment effects since I have reasonable concerns about the way the are calculated and there is no possible comparation with other EU countries.
Reviewer 2 Report
Accept as it is
Author Response
Dear Reviewer,
thank you for constructive suggestions which contributed to the improvement of the article quality.
Kind regards,
Authors
This manuscript is a resubmission of an earlier submission. The following is a list of the peer review reports and author responses from that submission.
Round 1
Reviewer 1 Report
After a brief comprobation for the case of output multipliers in Germany (2010), we have great concerns about the empirical results obtained in this investigation. In order to continue this review, the Excel files where calculations were maid is required.
Reviewer 2 Report
First of all, to discuss the transport industry's importance in economic growth, it seems essential to compare it with other industries. In this study, an international comparison of the transport industry is made, but no comparison between Croatia industries is made.
Furthermore, the methodology is quite orthodox and it is unclear where the originality of this study lies.
Reviewer 3 Report
This is an interesting study and its quality is good but it is not updated to the current policy dynamics since the I-O model is based on the data of 2003-2015 which is out of date.
The nature of these types of studies is static and therefore inferences are policy recommendations are too limited. The authors have provided a good comparative analysis of multipliers in Old and European countries with respect to Croatia in the transport sector but I could not see any novel contribution to the existing literature as it stands.
I would suggest that the authors should improve this research in the following ways:
- The authors use unnecessary references. It is more apparent in the first five sentences of the introduction section. Similarly, the authors cite multiple references in a sentence which is a poor academic practice, It should be avoided and no more than 3 references in a sentence are appropriate.
- The dynamic of form I-O analysis should be considered.
- The simulations in the transport sector could improve the quality of the manuscript